# Technical note: Fourier approach for estimating the thermal attributes of streams

Masahiro Ryo[1, 2], Marie Leys[1, 3], Christopher T. Robinson[1, 3]

[1]Department of Aquatic Ecology, EAWAG, 8600 Duebendorf, Switzerland, and Institute of Integrative Biology, ETH-Zürich,
8092 Zürich, Switzerland.
[2]Department of Civil Engineering, Tokyo Institute of Technology, 2-12-1 Ookayama, Meguro-ku, 152-0033 Tokyo, Japan
[3]Institute of Integrative Biology, ETH-Zürich, 8092 Zürich, Switzerland

*Correspondence to*: Masahiro Ryo (masahiroryo@gmail.com)

**Abstract.** Temperature models that directly predict ecologically-important thermal attributes across spatio-temporal scales are still poorly developed. This study developed an analytical method based on Fourier analysis to estimate seasonal and diel periodicities, as well as irregularities in stream temperature, at data-poor sites. The method extrapolates thermal attributes from highly resolved temperature data at a reference site to the data-poor sites on the assumption of spatial autocorrelation. We first quantified the thermal attributes of a glacier-fed stream in the Swiss Alps using 2-years of hourly-recorded temperature. Our
approach decomposed stream temperature as its average temperature of 3.8 °C, a diel periodicity of 4.9 °C, seasonal periodicity spanning 7.5 °C, and the remaining irregularity (variance) with an average of 0.0 °C but spanning 9.7 °C. These attributes were used to estimate thermal characteristics at upstream sites where temperatures were measured monthly, and we found that a diel periodicity and the variance strongly contributed to the variability at the sites. We evaluated the performance of our predictive mechanism, and found that our approach can reasonably estimate periodic components and extremes. We could also
estimate the variability in irregularity, which cannot be represented by other techniques that assume a linear relationship in temperature variabilities between sites. The results confirm that spatially extrapolating thermal attributes based on Fourier analysis can predict thermal characteristics at a data poor site. The R scripts used in this study are available in the Supplement.

## 1 Introduction

Temperature is a fundamental determinant of physical and biogeochemical patterns and processes in ecosystems. Organisms
respond to temperature via different adaptations and distinct life history strategies across various spatial and temporal dimensions (Cossins and Bowler, 1987). More broadly, global biodiversity has tracked external temperature transitions over millennia (Mayhew et al., 2012). In the Anthropocene, unprecedented rapid transitions in thermal regimes owing to changes in land use and climate are a global concern (IPCC, 2014). Consequently, adequate characterization of thermal regimes is an important prerequisite for understanding the role of temperature in ecosystems and for ecological conservation.

Thermal attributes (seasonal and diel periodicity patterns and irregular extremes), as representations of variability, have attracted a growing interest in recent climate-relevant ecological studies (e.g., Thompson et al., 2013). The combination of seasonal and diel periodicity cycles in temperature promotes diverse behaviors and spatial distributions of exothermic organisms according to respective life history strategies and life cycle stages (e.g., Vannote et al., 1980). In contrast, irregular
extremes such as heatwaves can induce physiological exposure and vulnerability (Paaijmans et al., 2013) as well as causing abrupt shifts in biogeochemical processes (Frank et al., 2015) and ecological assemblages (Thompson et al., 2013). Hence, the variability in thermal periodicities and irregular extremes need to be distinguished from average temperatures.

In freshwaters, ecological responses to thermal regime shifts may be less understood than in marine and terrestrial ecosystems (Thompson et al., 2013), even though freshwater biodiversity has experienced major decreases in recent decades
(WWF, 2014). Progress in understanding response patterns has been delayed partially because the quantification of thermal attributes is difficult for running waters. Physical disturbance in streams can be considerable due to periodic flow pulses (Poff et al., 1997). Particularly where flashy flows carry debris such as driftwoods or mobilize gravel riverbeds, e.g. braided rivers, the installation of data-loggers for long-term time-series monitoring can be logistically difficult. For natural floodplains acting as biodiversity hotspots (Tockner and Stanford, 2002), frequent time-series temperature data (daily or sub-daily) recorded
throughout a year or longer are difficult to acquire because the floodplain mosaic changes frequently and dramatically (Van der Nat et al., 2003). In these cases, researchers often rely on spot-measures of temperature or the use of reference temperature time-series at stations along the streamline. Both approaches have caveats (a lack of time-series information or likely bias in the data) when estimating the thermal attributes at a data-poor site, thereby limiting understanding of the ecological consequences in freshwaters. Estimating thermal stream attributes using both spot-measurements at study sites and time-series
measures at the nearest hydrological station would likely be a more robust approach.

There is still much room for improvement of statistical models for stream temperature (see review in Benyahya et al., 2007). Regression models employ correlative relationships with air temperature (e.g., Pilgrim et al., 1998) and streamflow (Webb et al., 2003), but correlative approaches considering water temperature at a nearby hydrological station along the streamline have not been implemented to date. Autoregressive models take into account the autocorrelation structure within
water temperature time-series data and also the correlation with external variables (Kothandaraman, 1971; Cluis, 1972; Long, 1972). To deal with seasonally changing parameters in regression and autoregressive models, periodic autoregressive models were introduced (Benyahya et al., 2007). However, temperature patterns at multiple temporal scales, particularly the combination of seasonal and diel periodicity patterns, are still rarely considered (Steel and Lange 2007). Moreover, temperature models that directly predict ecologically important thermal attributes as response variables do not exist presently, as many
models focus solely on temperature numeric values at a given time. Considerable error and bias in ecologically-relevant attributes can arise if they are calculated using a modeled time-series environmental condition (e.g., hydrologic indices calculated from a simulated river discharge; Ryo et al., 2015). Directly estimating ecologically-relevant thermal attributes, therefore, is required for reliably predicting associated ecological responses.

Fourier analysis (Fourier, 1878) is well-suited for analyzing combined multi-temporal patterns and predicting thermal attributes. Unfortunately, the use of Fourier analysis for assessing stream temperature patterns has slowed since its early emergence in 1970 (Kothandaraman, 1971; Cluis, 1972; Long, 1972; but see Maheu et al., 2015). This is surprising given the high potential for Fourier techniques to detect and describe periodicity at multiple scales in time-series data, and water temperature data in particular. Here, we investigate the application of Fourier analysis to the assessment of thermal patterns in running waters.

We developed an analytical method to estimating average, seasonal and diel periodicities in stream temperature, as well as irregular extremes at data-poor sites (i.e., spot measures) using Fourier analysis. We first quantified these thermal attributes with 2-years of hourly-recorded time-series temperature data from an alpine glacial-fed stream in Val Roseg, Switzerland (see Ward and Uehlinger (2003) for a synthesis of research conducted in this catchment). Using those results, we predicted thermal patterns at sites along the same stream, where monthly spot-measures of temperature were taken during the same 2-years. We compared the performance of the method with that of a linear regression model to underscore the advantages.

## 2 Methods

### 2.1 Compositional variables in stream temperature

We assume that hourly stream temperature $T(t)$ at a given time $t$ is composed of its long-term average value $\bar{T}$, a seasonal periodicity pattern $S(t)$, and diel periodicity pattern $D(t)$. The seasonal and diel periodicity patterns are driven by meteorological (e.g., solar radiation and precipitation) and hydrological (e.g., discharge and snow-/ice-melt) conditions. The remaining unexplained component of variance in the temperature time series results from multiple external factors such as sub-daily changes in weather conditions and a week-long heat waves: we call this component an irregularity $\varepsilon(t)$. A distinctive high/low value in the irregularity indicates thermal extremes that strongly disturb the periodicity patterns in temperature. Consequently, hourly stream temperature is expressed as:

$$T(t) = \bar{T} + S(t) + D(t) + \varepsilon(t) \tag{1}$$

### 2.2 Fourier analysis for temperature decomposition

Fourier transformation converts a function of time $T(t)$ into a function of frequency $G(f)$ by transforming time-series data into a sum of trigonometric curves. Both forward (2) and backward transformations (3) are identically reversible:

$$G(f) = \int_{-\infty}^{\infty} T(t)e^{-i2\pi ft}dt \tag{2}$$

$$T(t) = \int_{-\infty}^{\infty} G(f)e^{i2\pi ft}df \tag{3}$$

As measured stream temperature is a discrete variable, we used the fast Fourier transformation algorithm to perform the transforms. The algorithm searches for a single solution to identically explain a time-series variable $T(t)$ by summing the trigonometric curves ($e^{i2\pi ft}$) of different frequency $f$ amplified by a corresponding spectral intensity $G(f)$:

$$G(f) = \sum_{t=0}^{N-1} T(t)e^{-i2\pi ft}$$

(4)

$$T(t) = \frac{1}{N}\sum_{t=0}^{N-1} G(f)e^{i2\pi ft}$$

(5)

where $N$ is the length of the time series (rounded to the nearest power of 2), and equivalent to the number of summed curves.
As seen in Eq. (5), the time-series variable, hourly stream temperature $T(t)$, is formulated with summed trigonometric curves (with frequencies $f = 0, 1, 2, \ldots, N-1$) with magnitudes amplified by a corresponding spectral intensity $G(f)$. Frequencies with high spectral intensities are the most important contributors to variability of the time series. Stable patterns often dominate the variance of the time series, and result in strong spectral intensities at frequencies close to 24 hours and 1 year.

As the time-series temperature $T(t)$ is formulated by Eqs. (1) and (5), we consider that any of the terms in Eq. (1) results from a sum of a subset of the terms in Eq. (5). Although our intention is to explain time-series temperature with the four components above, Fourier analysis decomposes it into $N$ components based on a series of trigonometric curves. Terms in Eq. (5), therefore, need to be "classified" as belonging to the terms in Eq. (1). The mismatch in the number of components requires summing some of the trigonometric curves to best represent seasonal and diel periodicity patterns. However, it is unknown which curves at what frequencies and spectral intensities are required to sufficiently express these periodicity patterns. For selecting curves to be summed, we considered the condition that the seasonal and diel periodicities consist of their period lengths (inverse of frequency, $1/f$) within the ranges of 1–365 days and 1–24 hours, respectively. Moreover, we introduced a threshold value in spectral intensity for selecting only the dominant components as well as to avoid mixing noise with the periodicity patterns. The irregular component accounts for all frequencies with a spectral intensity below the (a) threshold value. Consequently, the shape of functions $S(t)$ and $D(t)$ depends on a spectral threshold value. We set the threshold value at 0.1 °C, a minimal unit of temperature measurement in the case study system (see sect. 2.4). The analysis was performed using the fast Fourier transform (*fft*) function of the "stats" library in R 3.1.2 (R Core Team, 2014).

## 2.3 Extrapolation to spot-measured temperature data

Assuming that thermal attributes are auto-correlated in space along the river continuum, we can extrapolate time-series data from reference locations to spot-measured data at sites along the same stream. For simplicity, we do not include external information (e.g., discharge and air temperature). After decomposing the time-series temperature $T_0$ as Eq. (1) at a reference site, the decomposed factors are used to estimate temperature $T_a$ at spot-measured site A along the same stream network. We assume that the four temperature components in Eq. (1) are linearly correlated between sites when other factors (e.g., major

tributary or groundwater inputs) affecting stream temperature along the network are low enough to maintain the spatial-autocorrelation temperature patterns between sites. The temperature at site A is formulated as:

$$T_a(t) = \bar{T}_a + S_a(t) + D_a(t) + \varepsilon_a(t) \tag{6}$$

$$= \beta_1 \bar{T}_0 + \beta_2 S_0(t) + \beta_3 D_0(t) + \beta_4 \varepsilon_0(t) + \beta_5 \tag{7}$$

where coefficients $\beta_{1-4}$ are the weighting parameters for each component ($> 0$), and $\beta_5$ is a parameter to adjust systematic bias. The following procedures are performed to estimate each parameter. First, linear regression between $T_a$ and $T_0$ corresponding

to temperatures at the measurement time of $T_a$ is conducted to estimate $\beta_1$ (slope) and $\beta_5$ (intercept). Second, $\beta_2$ and $\beta_3$ are estimated by minimizing the mean square error value based on the linear regression between an estimated $\beta_4 \varepsilon_0 = T_a - \{\beta_1 \bar{T}_0 + \beta_2 S_0(t) + \beta_3 D_0(t) + \beta_5\}$ and $\varepsilon_a$ at the corresponding measurement times. Third, $\beta_4$ is numerically estimated similarly as for the second step based on $T_a$ and the estimated $T_a$. Note that this approach will require a reasonably high density of spot measurements, covering the diel range—ideally including minimum and maximum—in different seasons.

To highlight the benefits of the extrapolation method, we compared the component extrapolation approach to a linear regression model that simply extrapolates time-series temperature based on the linear regression between $T_a$ and $T_0$. Importantly, if the coefficients $\beta_{1-4}$ have the same value as the slope of the linear regression and $\beta_5$ is the intercept, the Fourier approach is equivalent to the linear regression model. The R scripts used in this study are available in the Supplement.

## 2.4 The case study: an alpine glacier-fed stream (Val Roseg, Switzerland)

The method was applied in the Roseg catchment, an alpine valley located in the Bernina massif of the Swiss Alps (Fig. 1). The catchment area is 66.5 km$^2$ and ca. 30% glaciated (Swiss National Hydrological and Geological Survey (OFEV); year of record 2010). Elevations range from 1766 to 4049 m a.s.l. The Roseg River is fed by meltwaters of the Tschierva and Roseg glaciers. The Roseg glacial runoff first drains into the pro-glacial lake Roseg before merging with the flow from the Tschierva glacier (Fig. 1). The thermal attributes of the glacial meltwaters and runoff from the lake strongly influence the seasonal and diel

periodic thermal patterns in this river. Mean annual discharge at the end of the catchment was 2.8 m$^3$ s$^{-1}$ (discharge record averaged for 1955–2013) and daily discharge ranged from 3.3 to 19.6 m$^3$ s$^{-1}$ in July and August and from 0.2 to 2.0 m$^3$ s$^{-1}$ between November and March 2013 (OFEV). The study system comprises (i) a long proglacial reach below the Tschierva glacier, exhibiting extremely low temperatures due to glacial runoff (kryal), (ii) a single-thread channel downstream of the confluence of the proglacial reach and the Roseg lake outlet, (iii) a complex braided floodplain, and (iv) a constrained reach

extending to the end of the catchment where the Pontresina hydrological station is located (see Tockner et al., 1997; Uehlinger et al., 2003 for a detailed description).

Hourly time-series temperature was recorded at a reference site, the Pontresina hydrological station (site R in Fig. 1: 46°29'23.6"N, 09°53'53.3"E; 1766 m a.s.l.) in 2012 and 2013 (Fig. 2a: provided by OFEV). Spot-measured water temperature was taken monthly at two sites, one located within the proglacial reach (site A: 46°24'38.3"N, 09°51'31.2"E; 2106 m a.s.l.)

and one below the lake outlet confluence reach (site B: 46°25'05.9"N, 09°51'27.1"E; 2054 m a.s.l.) (Fig. 1). For both sites,

stream temperature was measured using a conductivity meter (WTW LF323, Weilheim, Germany) at different daily times on each visit from April to October 2012 and 2013 (in total 14 times).

## 3 Results

### 3.1 Thermal attributes at the reference site

By converting the time-series temperature data at the reference site R (Fig. 2a) to the frequency domain, we identified frequency ranges with high spectral power (Fig. 3). Nine trigonometric curves exceeded the threshold of 0.1°C (Table 1). Based on period length (Table 1), these curves were allocated to seasonal and diel components: three curves for seasonal periodicity patterns (Fig. 2b); and six curves for diel periodicity patterns (Fig 2c). The time-series temperature at the reference site R (Fig. 2a) was thus decomposed into an average of 3.8 °C, a seasonal cycle spanning 7.5 °C (Fig. 2b), a diel cycle

spanning 4.9 °C (Fig. 2c), and an irregularity spanning 9.7 °C with an average of 0.0 °C and standard deviation of 0.92 °C (Fig. 2d). Hourly stream temperatures excluding the irregularity (i.e., $\bar{T} + S(t) + D(t)$; Fig. 2e) explained 92% of the data variability ($r^2$), indicating a successful decomposition of the time-series data and a high reliability of the approach to characterize stream temperature components at the reference site. The irregularity had a normal distribution (no inferred bias), indicating that the seasonal and diel periodicity patterns were extracted accurately from the original time-series data.

### 3.2 Estimating time-series temperature for spot-measured stream sites

Temperature at site A was approximately twofold lower than at site B during the study period. Temperature at site A had an average of 2.3 °C and spanned 0.4–4.8 °C; temperature at site B had an average of 5.4 °C and spanned 0.8–9.2°C. Temperatures at site A and B were linearly correlated to temperature at the reference site R ($r^2$ = 0.60 and 0.92, respectively; Fig. 4). This result indicates that spatial autocorrelation of the thermal attributes exists between them, even though a thermal influence from

the pro-glacial lake differentiates the thermal patterns between sites A and B (Fig. 1).

For Eq. (7), the proposed method estimated that the weighting parameters at site A ($\beta_{1-4}$ = {0.32, 0.32, 0.50, 0.70}) showed higher heterogeneity in the relative contributions of the components than those at site B ($\beta_{1-4}$ = {0.97, 0.97, 0.99, 0.93}). For site A, the relative contribution of diel variability and irregularity to the estimated temperature ($\beta_3$ = 0.50, $\beta_4$ = 0.70) was higher than the average and seasonal variability ($\beta_1 = \beta_2$ = 0.32). This result indicates that our approach can accurately estimate

periodic components and extremes, including the variability in irregularity that cannot be represented by linear regression focusing on an average estimate. At site B, the parameter composition was less variable (0.93–0.99), and therefore the performance of the decomposition approach was similar to the performance of the linear model. Using the developed method, the temperature at site A was estimated better than the estimated temperature using linear regression ($r^2$ = 0.66 and 0.60, respectively). For site B, both approaches performed equivalently ($r^2$ = 0.92). These differences between sites A and B indicate

that the thermal attributes at site A are more different from the reference site R than those at site B. The thermal attributes at site A is not affected by a thermal effluence from the pro-glacial lake (Fig. 1).

Based on the obtained parameters, time-series temperatures at sites A and B were inferred (Fig. 5). At site A, due to relatively high contributions of diel cycles and irregularities ($\beta_3 = 0.50$, $\beta_4 = 0.70$), higher hourly variability was estimated by the proposed approach than the linear regression (Fig. 5a). The maximum values estimated by the proposed approach and linear regression were clearly distinct at 5.9 °C and 4.0 °C, respectively, while an average of 1.1 °C was equivalent for both estimates. The

linear model clearly underestimated (< 4.0°C) the high temperatures recorded at 4.4 °C on 30 May 2012 and 4.8 °C on 5 June 2013. In contrast, the proposed approach inferred a possibility of temperature reaching 5.9 °C, having a 97.5% percentile value of 4.0 °C. This difference between approaches indicates the ability of the proposed approach to estimate extreme thermal pulses and their occurrence probability.

## 4 Discussion

Currently, no stream temperature models explicitly predict ecologically important thermal attributes (seasonal and diel periodicity patterns and irregular extremes) because of the difficulty in capturing the combined patterns at multiple temporal scales in stream environments (Benyahya et al., 2007). This study developed a regression approach that predicts these thermal attributes at data-poor sites based on the pre-analysis of time-series temperature data at a data-rich reference site along the stream. The method merges a Fourier transformation technique into a linear regression model to better represent periodic

patterns at multiple temporal scales. The approach could estimate the relative composition of thermal attributes from a limited number of spot-measured data (see Eq. 7), while linear regression weighted the composition equally. The results emphasized the significance to develop further ecological-based thermal prediction models, aiming at deeper understandings in ecological responses to thermal attributes (Paaijmans et al., 2013; Thompson et al., 2013; Frank et al., 2015).

The developed prediction method confirmed its potential to evaluate the relative contribution of thermal attributes at a data-

poor site. The method is somewhat comparable with multiple linear regressions ($Y = \beta_0 + \beta_1 X_1 + \dots$) in terms of the assumption that the thermal attributes are independent of each other and auto-correlated between sites, and their relative contributions change according to location. The unique difference is that the developed method can directly use the periodicity patterns and irregularities for prediction as Eq. (7). This type of model was not addressed in a recent review on temperature models (Benyahya et al., 2007). As the statistical expression of our approach is linear (Eqs. 1 and 6), it can be easily coupled with

approaches in the review; i.e., using other regression models employing air temperature (e.g., Pilgrim et al., 1998) and streamflow (Webb et al., 2003). Adding such information has the potential to increase accuracy, especially if these factors contain unique information that is unexplained by the spatial correlation of water temperature between sites. For example, if discharge represents a volume of snowmelt water that can influence the correlative relationship of water temperature between sites, inclusion of discharge into the model's structure would increase accuracy.

The advantage of Fourier decomposition analysis at multiple temporal scales is the ability to evaluate the possible range maxima for each thermal attribute of time-series data (Fig. 2). Fourier analysis clearly detected strong periodicity patterns in time-series temperature data (Fig. 3) and represented them with a small number of trigonometric curves (Table 1). By doing

so, we found that the irregularity had broader range (9.7 °C) than the spanning ranges of seasonal and diel periodicity patterns (7.5 °C and 4.9 °C, respectively) at the study stream. This high irregularity may be an important thermal attribute in glacier-fed streams. Further, our results clearly showed that seasonal trends existed in the amplitudes of diel periodicities and irregularities, both having lower variability in winter and greater variability in spring (Figs. 2c-d) (Hopkins, 1971). Importantly, these characteristics are difficult to detect without coupling the periodicity patterns at multiple temporal scales. By decomposing the time-series temperature data at the reference site as a pre-analysis for prediction, this decomposition method (see sect. 2.1 and 2.2) can be useful for quantifying thermal attributes in ecological studies.

The proposed method is highly promising towards evaluating potential changes in thermal variability due to climate change or anthropogenic thermal effluents in rivers and streams (Caissie, 2006; Webb et al., 2008). For instance, environmental change could modify each component of stream temperature with a different degree of severity such as increasing the average, shifting the peak timing in seasonality, reducing diel variability, or amplifying the irregularity. In addition, as thermal regimes play a key role in aquatic ecosystem structure and functioning (e.g., environmental niches determining community composition, biogeochemical cycles, Vannote et al., 1980), each component of stream temperature can be used to better understand ecosystem dynamics and responses (Thompson et al., 2013).

While demonstrating the broad applicability of this approach, we caution over some of its limitations. First, the analysis connecting Eqs. (1) and (7) is threshold dependent. We first estimated an appropriate range of threshold value visually so as to capture a handful of trigonometric curves (i.e., the three major peaks were shown in Figure 3). Then, we compared the patterns modeled based on some threshold values. Too low threshold value results in high sensitivity to noise, while too high threshold value results in high insensitivity to periodic patterns. We compared the performance of the models based on a set of threshold values (0.05, 0.1, and 0.2 °C etc.) and determined graphically as the value 0.1 °C clearly separated periodicity patterns and irregularities (Figure S1; supplement). Therefore, a threshold value must be carefully chosen and needs to be evaluated whether the irregularity attribute is unbiased. Second, although the concept is feasible, complete validation was limited due to the low number of sample sites. To obtain more robust estimation, the model needs additional validation as well as calibration. Third, if a systematic shift in thermal pattern is observed during a target period, the method would need adjustment. Our target 2-year period was short enough to neglect long-term trends in temperature. For example, some studies using long-term records (>20 years) detected an inter-annual increasing trend in mean temperature due to anthropogenic thermal discharge and land-use change (Beschta and Taylor, 1988; Hostetler, 1991). In such cases, the proposed equation in Eq. (1) must be modified because it assumes constant average temperature over a target period for analysis; i.e., $\bar{T}$ should be a function of time as $\bar{T}(t)$. Last, the assumption of linearly-related components between the reference site and other sites is not always met in all cases. For instance, glacier-fed and groundwater-fed streams can be composed of different seasonal and diel periodicity patterns (Brown and Hannah, 2008). In our target system, flow from the Roseg pro-glacial lake differentiated thermal patterns of sites A and B (Fig. 1). Even though our analysis confirmed that the linear assumption was apparent in this system ($r^2 = 0.60$; Fig. 4), incorporation of the thermal pattern at the lake outlet into the analysis would certainly increase estimation accuracy.

*Acknowledgements.* The study was done as a fellow in the "Young Researchers Exchange Program between Japan and Switzerland" under the "Japanese-Swiss Science and Technology Programme" (EG 11-2015) and supported by the Japan Society for the Promotion of Science (26-11771). The data used in this study were collected for a Swiss National Science

Foundation project (31003A_152815). We thank the Swiss National Hydrological and Geological Survey from the Federal Office for the Environment (OFEV) for kindly providing water temperature data and S. Blaser for field assistance. We are indebted to the anonymous reviewers for their constructive reviews and comments. We thank India Mansour for proofing language.

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

**Table 1:** Frequency $f$, period $1/f$, and spectral intensity $G(f)$ of trigonometric curves (threshold of $G(f) > 0.1$) composing the thermal attributes: average $\overline{T}$, seasonal periodicity pattern $S(t)$, and diel periodicity pattern $D(t)$.

| Attribute | $f$ [Hz] | $1/f$ [hrs] | [days] | $G(f)$ [°C] |
|---|---|---|---|---|
| $\overline{T}$ | 0 | – | | 3.78 |
| $S(t)$ | 2 | 8760 | 365 | 2.01 |
| | 4 | 4380 | 183 | 0.11 |
| | 6 | 2920 | 122 | 0.19 |
| $D(t)$ | 728 | 24.1 | 1 | 0.27 |
| | 730 | 24 | 1 | 0.59 |
| | 732 | 23.9 | 1 | 0.28 |
| | 734 | 23.9 | 1 | 0.10 |
| | 1458 | 12.0 | 0.5 | 0.12 |
| | 1460 | 12 | 0.5 | 0.21 |

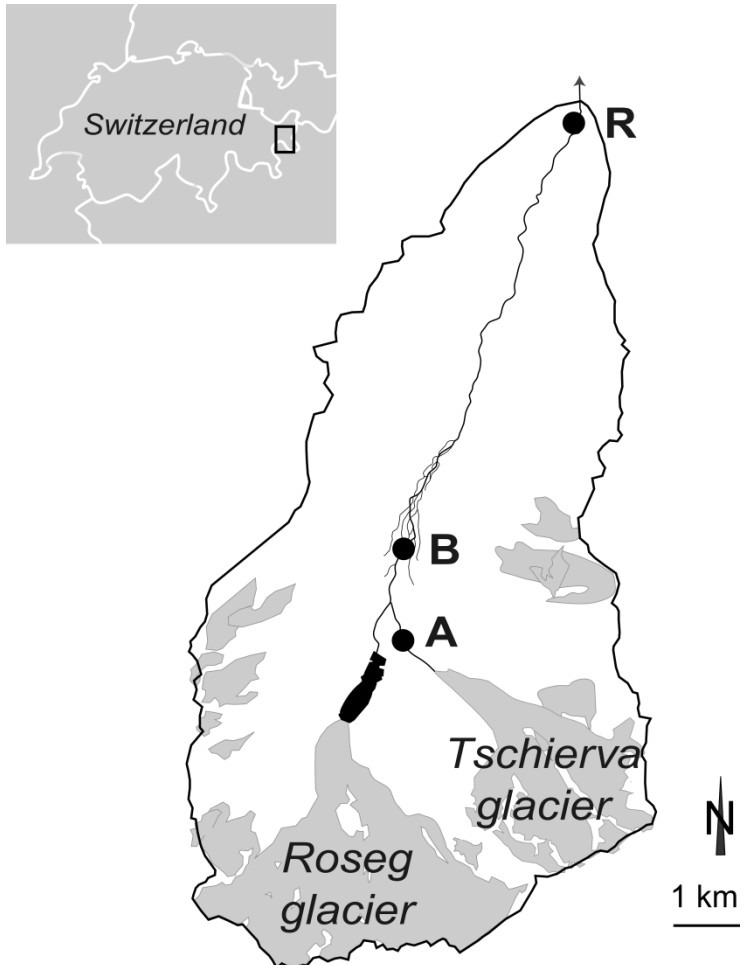

**Figure 1:** Location of the Val Roseg catchment in Switzerland. Study and reference (i.e., Pontresina hydrological station) sites
are indicated by black dots and labeled A, B and R, respectively. Glaciers are shown as shaded areas with the pro-glacial lake
shown below the Roseg glacier (modified from Uehlinger et al. 2003).

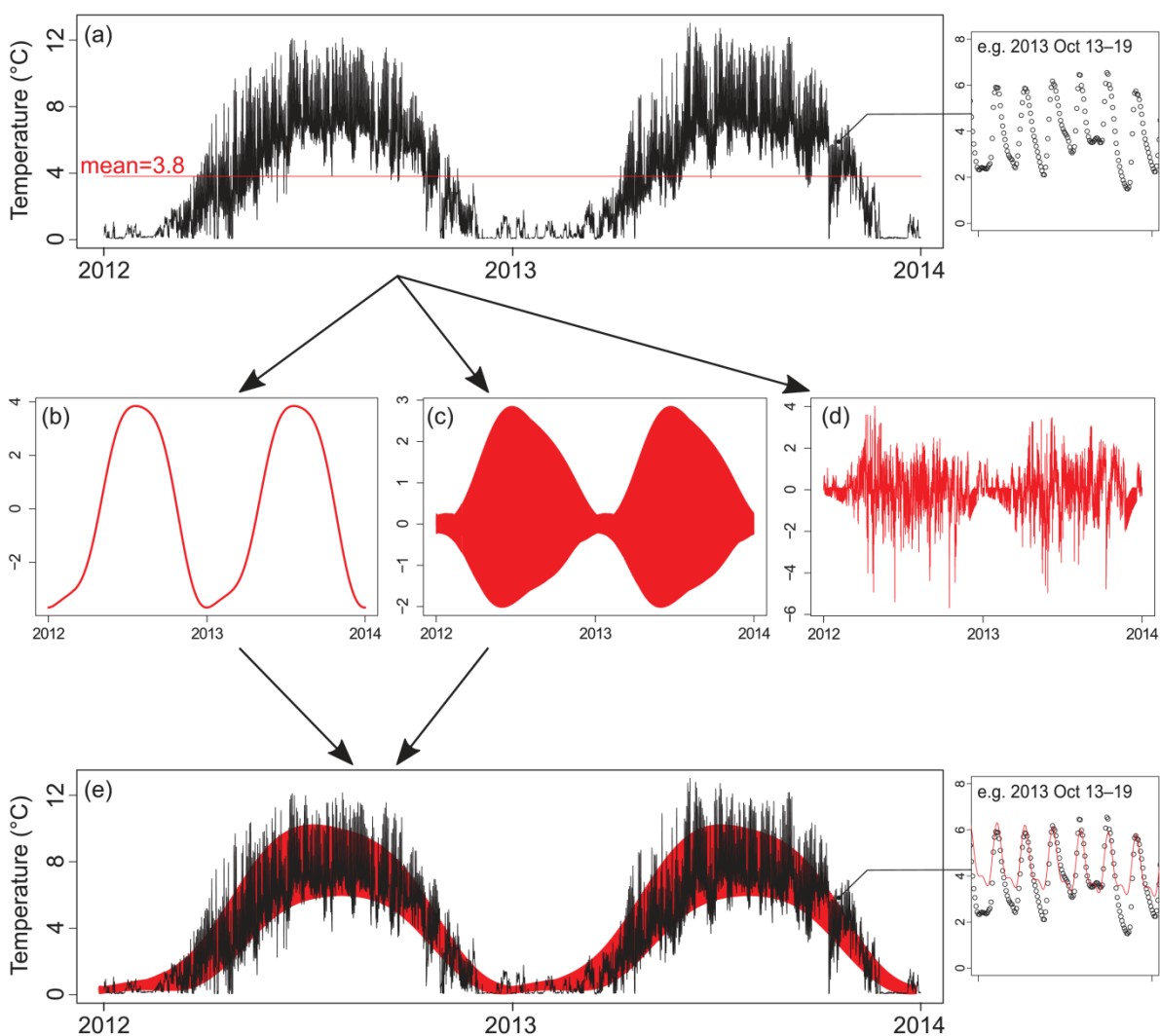

**Figure 2:** Hourly time-series temperature at the reference site decomposed using Fourier decomposition analysis. (a) The observed record $T_a$ in 2012–2013 is decomposed into an average value (a), seasonal periodicity pattern $S_a$ (b), diel periodicity pattern $D_a$ (c), and the irregularity $\varepsilon_a$ (d). The comparison between the observed and measured temperature excluding $\varepsilon_a$ at the reference site (e).

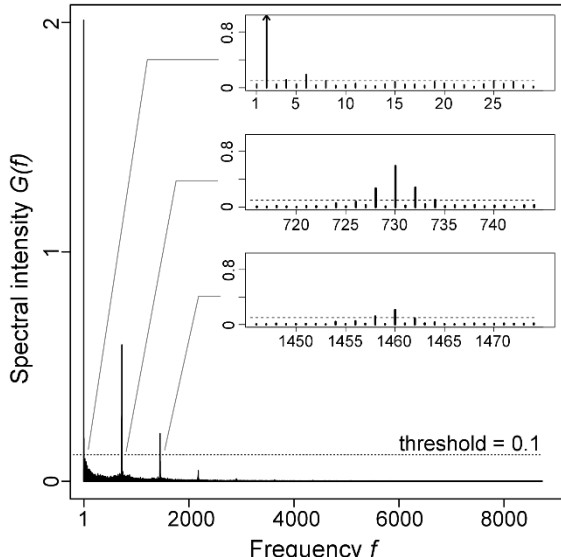

**Figure 3:** Frequency and spectral intensity of trigonometric curves converted from the 2-year temperature data at the reference site. Trigonometric curves whose spectral intensity was higher than the threshold value are found in Table 1.

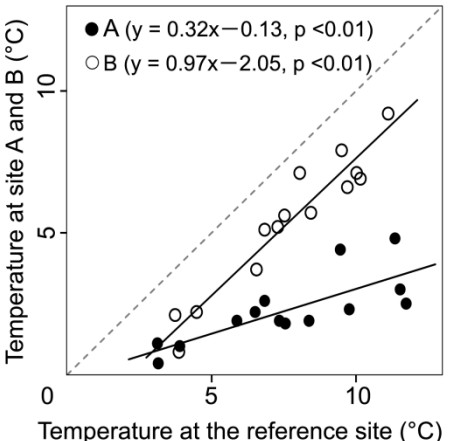

10  **Figure 4:** Linear regression of stream temperature at site A and B with the reference site temperature R.

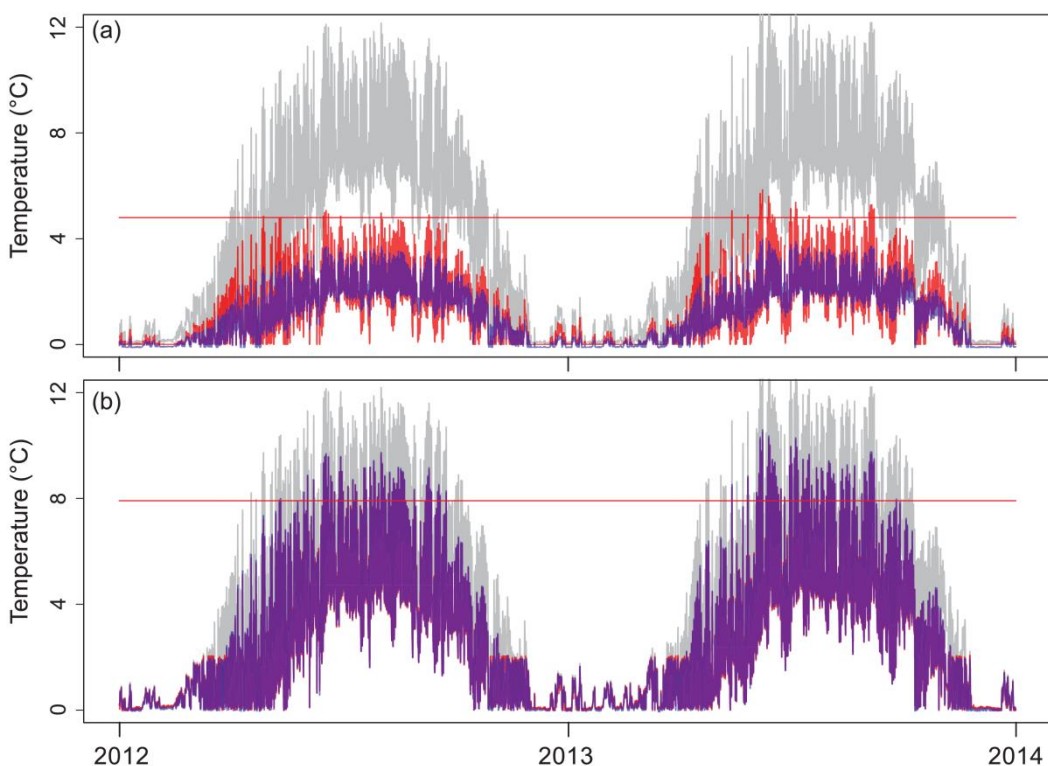

**Figure 5:** Hourly stream temperatures estimated using Fourier decomposition (red) and linear regression (purple) at site A and B (plots a and b, respectively) and temperature at the reference site (grey). Horizontal red line represents the maximum temperature recorded from the monthly spot measures (4.8 °C on 5 June 2013 at site A, and 7.9 °C on 9 July 2013 at site B).