# Peer review of "Technical note: Fourier approach for estimating the thermal attributes of streams"

_Hydrology and Earth System Sciences, 2016_

## Referee Comment (RC1) · Anonymous Referee #1 · 14 Jun 2016

This manuscript presents a new method that predicts stream temperature at data-poor sites using thermal attributes (e.g., diel, seasonal periodicities and irregularities) from Fourier analysis of a reference site, assuming spatial auto-correlation between sites along the river continuum. Its contribution to understanding stream temperatures lies in its contrast from linear regressions: it provides site-specific weighing of these reference site thermal attributes to infer stochastic behaviors at data-poor sites. The authors present, in the introduction, a solid argument for closing the current gap in research surrounding understanding of stream temperatures, and the proposed method of Fourier analysis in determining stream thermal attributes is very clear. However, the role of the proposed method in filling this gap may need some rewording. I would recommend this article for publications with some edits.

Below is a list of scientific questions and suggestions that arose while reading the

manuscript:

1. Page 2, line 5–6: "Progress in understanding response patterns have been delayed partially because the quantification of thermal attributes is difficult for running waters."
   The introduction stresses that understanding of thermal attributes in lotic ecosystems is hindered by scarce temporal observations, which certainly is a limitation in current research. However, this method hinges on a reference hydrologic station (i.e. a site that is not data-scarce) to elucidate thermal attributes at a data-scarce site. The manuscript's current focus seems to lie in extrapolating within a basin where there is abundant data from at least one location, and it may be helpful to state this explicitly because the description of the limitations in currently available methods/research that are mentioned in the introduction may signal to readers that the proposed method will be rooted solely in spot measurements.

2. Page 3, line 5: The authors consider external factors such as weather conditions as part of the irregularity component. However, in other climates, these meteorological conditions would likely factor into the diel (sun) and seasonal (rain) periodicity patterns as well.

3. Page 6, line 5: Are the $r^2$ values (0.66 vs. 0.6) significantly different to state that the proposed method works better than linear regression? It might make more sense to first stress that the method can successfully recreate extremes (cf. linear regression), and then state that it is better at estimating temperatures at site A.

4. Page 7, line 16: Is the threshold value specific to the system in question? How was this value chosen, and what implications might this have?

Below is a list of technical corrections:

[Figure]

1. "Thermal attributes" should be explicitly described earlier in the introduction. The full description is currently on page 2, line 21, but "thermal attributes" are first mentioned well before this line.

2. Page 2, line 26–27: "...multiple scales in time-series data, **and** water temperature in particular."

3. Page 2, line 33: "...sites **along** the same stream..."

---

## Referee Comment (RC2) · Anonymous Referee #2 · 21 Jun 2016

Key Points:

Temperature is an important determinant of physical and biogeochemical patterns and processes in ecosystems. Temperature models that predict "thermal attributes" (diel and seasonal periodicity and irregular extremes) at sites with spot temperature measurements are poorly developed. The authors present a new temperature modelling method based on Fourier analysis to determine these "thermal attributes" at data poor sites. The authors' method is dependent on having highly resolved temperature data at a reference site and on the assumption that there is spatial autocorrelation between the reference site and data poor site. The authors' model performs similarly to a linear regression method with the exception that the authors' model is better equipped at recognizing extreme thermal pulses and their probability. I recommend this manuscript for publication with edits.

[Figure]

Scientific questions and suggestions:

Page 1, line 16-17: "The results confirm that the developed method can infer stochastic behaviors in stream thermal attributes at spot-measured sites." It would be beneficial to the reader to reword this sentence so that it reflects the requirement of having highly resolved temperature data at a reference site and the assumption of spatial autocorrelation between the reference site and data poor site that this method relies on.

Page 2, line 3-13: The introduction correctly stresses the importance of knowing "thermal attributes" at a given site with regards to an ecosystem. The authors go on to describe that determining "thermal attributes" can be difficult and unrealistic because of the need for highly resolved temperature data. They present a strong argument for the need for improved modeling that can rely on sparsely collected temperature data. The introduction makes it sound as if the temperature modelling method presented in this manuscript does just that. However, the authors' model is dependent on having two years' worth of hourly temperature data at a reference site. In addition, it relies on the assumption that there is spatial autocorrelation between the reference site and the data poor site. It would be beneficial to reword the introduction so that this information is more explicit.

Page 4, line 9-10: The authors do not include discharge and air temperature data in their methods for simplicity. Would adding this information to the Fourier analysis method improve its performance when compared to the linear regression method?

Page 6, line 3-5: It appears that the method performs comparably to a linear regression with the exception that the presented method captures extreme thermal pulses and their probability. The linear regression method does not do this. It would be beneficial to emphasize this result and include it in the Abstract.

Technical, spelling, and grammatical edits:

Page 1, line 10: It would be beneficial to define explicitly what "thermal attributes" are

earlier in the manuscript. The authors do so on Page 2, line 21-22. However, the term is used several instances before this definition.

Page 1, line 11: "Based on Fourier analysis, this study developed. . ." Misplaced modifier

Page 1, line 12-13: "We first quantified. . .Stream temperature was accurately decomposed. . ." The first sentence is active voice while the second sentence is passive voice. The introduction should remain in active voice.

Page 2, line 5: Progress in understanding response patterns has been delayed. . ." Subject verb agreement

---

## Author Comment (AC1) · 21 Jun 2016

Reply to RC1

—————————

1. Page 2, line 5–6: "Progress in understanding response patterns have been delayed partially because the quantification of thermal attributes is difficult for running waters." The introduction stresses that understanding of thermal attributes in lotic ecosystems is hindered by scarce temporal observations, which certainly is a limitation in current research. However, this method hinges on a reference hydrologic station (i.e. a site that is not data-scarce) to elucidate thermal attributes at a datascarce site. The manuscript's current focus seems to lie in extrapolating within a basin where there is abundant data from at least one location, and it may be helpful to state this explicitly because the de-

scription of the limitations in currently available methods/research that are mentioned in the introduction may signal to readers that the proposed method will be rooted solely in spot measurements.

—[Our reply]—

As the reviewer has clarified, our approach for temperature estimation based on extrapolation cannot be applied where no monitoring station exists along the streamline. To avoid misleading readers, we would have modified the last two sentences in this paragraph and a sentence in the next paragraph according to the reviewer's comment as follows:

Before (p.2, line 10-): Often in these cases, researchers rely on spot-measures of temperature at study sites and thus lack time-series temperature, thereby limiting understanding of the ecological consequences of thermal attributes in freshwaters. Clearly, an estimate of the thermal attributes at spot-measured sites would benefit this understanding.

After: Often in these cases, researchers rely on spot-measures of temperature at study sites although lacking time-series temperature or, otherwise refer to temperature time-series monitored at a near hydrological station along the streamline although likely being biased in thermal attributes. Either one has the problem, thereby limiting understanding of the ecological consequences in freshwaters. Estimating thermal stream attributes from both spot-measurements at study sites and time-series at the nearest hydrological station would allow more robust estimates.

Before (p.2, line 15): For instance, regression models employ correlative relationships with air temperature (e.g., Pilgrim et al., 1998) and streamflow (Webb et al., 2003).

After: For instance, regression models employ correlative relationships with air temperature (e.g., Pilgrim et al., 1998) and streamflow (Webb et al., 2003), while a correlative approach considering water temperature at a near hydrological station along the

streamline have not been implemented yet.

————————————

2. Page 3, line 5: The authors consider external factors such as weather conditions as part of the irregularity component. However, in other climates, these meteorological conditions would likely factor into the diel (sun) and seasonal (rain) periodicity patterns as well. For better clarification we would have modified the description as follows:

—[Our reply]—

We assume that hourly stream temperature T(t) at a given time t is generally composed of the average value T ÌĚ, seasonal periodicity pattern S(t), and diel periodicity pattern D(t). The seasonal and diel periodicity patterns can be driven by meteorological (e.g., solar radiation and precipitation) and hydrological (e.g., discharge and snow-/ice-melt) conditions. The remaining component, which is unexplained by these three components, results from multiple external factors such as stochastic sub-daily changes in weather conditions (sunshine, rain, wind, etc.): we call this component an irregularity $\varepsilon$(t).

Before (p.3, line 4): We assume that hourly stream temperature T(t) at a given time t is generally composed of the average value T ÌĚ, seasonal periodicity pattern S(t), and diel periodicity pattern D(t). The remaining component, which is unexplained by these three components, results from multiple external factors such as weather conditions (sunshine, rain, wind, etc.): we call this component an irregularity $\varepsilon$(t).

After: We assume that hourly stream temperature T(t) at a given time t is generally composed of the average value T ÌĚ, seasonal periodicity pattern S(t), and diel periodicity pattern D(t). The seasonal and diel periodicity patterns can be driven by meteorological (e.g., solar radiation and precipitation) and hydrological (e.g., discharge and snow-/ice-melt) conditions. The remaining component, which is unexplained by these three components, results from multiple external factors such as stochastic sub-daily

changes in weather conditions: we call this component an irregularity $\varepsilon$(t).
* * *
3. Page 6, line 5: Are the r2 values (0.66 vs. 0.6) significantly different to state that the proposed method works better than linear regression? It might make more sense to first stress that the method can successfully recreate extremes (cf. linear regression), and then state that it is better at estimating temperatures at site A.

—[Our reply]—

We fully agree with the reviewer's suggestion. We would have modified the text to better stress out the success in our primary aim recreating extremes: "This result indicates that our approach can accurately estimate periodic components and extremes including the variability in irregularity, that cannot be represented by linear regression focusing on an average estimate." Inserted in p. 5 line 2 just before the sentence starting from "At site B, . . ."
* * *
4. Page 7, line 16: Is the threshold value specific to the system in question? How was this value chosen, and what implications might this have?

—[Our reply]—

Yes, it would be system-dependent. The paragraph would have been modified accordingly to clearly indicate how we selected the threshold (p. 7, line 15). See also new supplement figure S1:

First, the analysis connecting Eqs. (1) and (7) is threshold dependent. We firstly estimated an appropriate range of threshold value visually so as to capture a handful of trigonometric curves (i.e., the three major peaks were focused in Figure 3). Then, we compared the patterns modeled based on some threshold values. Too low threshold value results in too sensitive to noises, while too high threshold value results in too insensitive to periodic patterns. We compared the performance of the models based on a set of threshold values (0.05, 0.1, and 0.2°C etc.) and determined graphically as the value 0.1°C clearly separated periodicity patterns and irregularities (Figure S1). Therefore, a threshold value must be carefully chosen and needs to be evaluated whether the irregularity attribute is unbiased.

————————

5. Below is a list of technical corrections: 1. "Thermal attributes" should be explicitly described earlier in the introduction. The full description is currently on page 2, line 21, but "thermal attributes" are first mentioned well before this line. 2. Page 2, line 26–27: "...multiple scales in time-series data, and water temperature in particular." 3. Page 2, line 33: "...sites along the same stream..."

—[Our reply]—

We would have modified technical corrections 1–3 suggested by the reviewer accordingly.

————————

Please also note the supplement to this comment:
http://www.hydrol-earth-syst-sci-discuss.net/hess-2016-238/hess-2016-238-AC1-supplement.pdf

————————————————————

[Figure]

**Fig. 1.** Figure S1 The relationships of the threshold selection to extracted seasonal and diel periodic patterns in temperature (also see Figure 2)

**Supplement:**

**(a) Threshold = 0.05 (too sensitive)**

[Figure]

[Figure]

**(b) Threshold = 0.1 (used in this study)**

[Figure]

[Figure]

**(c) Threshold = 0.2 (over simplified)**

---

## Referee Comment (RC3) · Anonymous Referee #1 · 23 Jun 2016

The authors have done a good job in responding to my comments on their initial version of the manuscript; thank you for the detailed response. The new version of the manuscript has largely addressed my concerns, and as I stated in my initial review, I feel that the manuscript documents a novel and clear method that fills a current gap in research surrounding understanding of stream thermal attributes. I have a minor comment that could help the overall readability of the manuscript, which should be easy to address. Following this modification, I would support the publication of the manuscript in HESS.

Please note that line numbers refer to previous numbering:

1. Page 2, line 10: To which "problem" are you specifically referring? You could edit this section to read: "...near a hydrological station along the streamline, which

is likely biased in thermal attributes. Both limit the understanding of ecological consequences in freshwater."

---

## Author Comment (AC2) · 28 Jun 2016

We appreciate your help to further increase the readability.

1. Page 2, line 10: To which "problem" are you specifically referring? You could edit this section to read: "...near a hydrological station along the streamline, which is likely biased in thermal attributes. Both limit the understanding of ecological consequences in freshwater."

—[Our reply]—

We would revise the section as the following:

Before (p.2, line 10-): Often in these cases, researchers rely on spot-measures of temperature at study sites and thus lack time-series temperature, thereby limiting understanding of the ecological consequences of thermal attributes in freshwaters. Clearly, an estimate of the thermal attributes at spot-measured sites would benefit this under-standing.

After: Often in these cases, researchers rely on spot-measures of temperature at study sites lacking time-series temperature or refer to temperature time-series monitored at a nearby hydrological station along the streamline, although likely being biased in thermal attributes. Both datasets have caveats (a lack of time-series or bias in data) when estimating the thermal attributes at a data-poor site, thereby limiting understanding of the ecological consequences in freshwaters. Regardless, estimating thermal attributes from both spot-measurements at study sites and time-series at the nearest hydrological station would allow more robust estimates.

---

## Author Comment (AC3) · 28 Jun 2016

1. Page 1, line 16-17: "The results confirm that the developed method can infer stochastic behaviors in stream thermal attributes at spot-measured sites." It would be beneficial to the reader to reword this sentence so that it reflects the requirement of having highly resolved temperature data at a reference site and the assumption of spatial autocorrelation between the reference site and data poor site that this method relies on.

—[Our reply]—

As the reviewer has suggested, we would modify the sentence as follows:

Before (p.1, line 16): The results confirm that the developed method can infer stochastic

[Figure]

behaviors in stream thermal attributes at spot-measured sites.

After: The results confirm that the developed method, spatially extrapolating thermal attributes based on Fourier analysis, can infer stochastic behaviors in stream thermal attributes at a data poor site.

—

In addition, to increase readability, we described the method more precisely in abstract:

Before (p.1, line 11): This study developed an analytical method to estimate seasonal and diel periodicities as well as irregularities in stream temperature at data-poor sites based on Fourier analysis.

After: This study developed an analytical method to estimate seasonal and diel periodicities as well as irregularities in stream temperature at data-poor sites based on Fourier analysis extrapolating thermal attributes from highly resolved temperature data at a reference site, on the assumption of spatial autocorrelation.
* * *
2. Page 2, line 3-13: The introduction correctly stresses the importance of knowing "thermal attributes" at a given site with regards to an ecosystem. The authors go on to describe that determining "thermal attributes" can be difficult and unrealistic because of the need for highly resolved temperature data. They present a strong argument for the need for improved modeling that can rely on sparsely collected temperature data. The introduction makes it sound as if the temperature modelling method presented in this manuscript does just that. However, the authors' model is dependent on having two years' worth of hourly temperature data at a reference site. In addition, it relies on the assumption that there is spatial autocorrelation between the reference site and the data poor site. It would be beneficial to reword the introduction so that this information is more explicit.

—[Our reply]—

As another reviewer has also pointed out the lack of explanation which can mislead readers, we would modify the last two sentences in this paragraph and a sentence in the next paragraph:

Before (p.2, line 10-): Often in these cases, researchers rely on spot-measures of temperature at study sites and thus lack time-series temperature, thereby limiting understanding of the ecological consequences of thermal attributes in freshwaters. Clearly, an estimate of the thermal attributes at spot-measured sites would benefit this understanding.

After: Often in these cases, researchers rely on spot-measures of temperature at study sites lacking time-series temperature or refer to temperature time-series monitored at a nearby hydrological station along the streamline, although likely being biased in thermal attributes. Both datasets have caveats (a lack of time-series or bias in data) when estimating the thermal attributes at a data-poor site, thereby limiting understanding of the ecological consequences in freshwaters. Regardless, estimating thermal attributes from both spot-measurements at study sites and time-series at the nearest hydrological station would allow more robust estimates.

—-

Before (p.2, line 15): For instance, regression models employ correlative relationships with air temperature (e.g., Pilgrim et al., 1998) and streamflow (Webb et al., 2003).

After: For instance, regression models employ correlative relationships with air temperature (e.g., Pilgrim et al., 1998) and streamflow (Webb et al., 2003), whereas a correlative approach considering water temperature at a nearby hydrological station along the streamline has not been implemented yet.

————————————————————————————————————————

3. Page 4, line 9-10: The authors do not include discharge and air temperature data in their methods for simplicity. Would adding this information to the Fourier analysis

method improve its performance when compared to the linear regression method?

—[Our reply]—

Yes, adding information on discharge and air temperature has a potential to increase accuracy, especially if these factors contain unique information which is unexplained by the spatial correlation of water temperatures between sites. For example, if discharge can represent a volume of snow-melting water that may influence the correlative relationship of water temperature between sites, the inclusion of discharge into the model's structure would increase the accuracy.

We would include this point in discussion:

(After p.6, lines 28–29): This type of model was not addressed in a recent review on temperature models (Benyahya et al., 2007). As the statistical expression of our approach is linear (Eqs. 1 and 6), it can be easily coupled with approaches in the review; i.e., using other regression models employing air temperature (e.g., Pilgrim et al., 1998) and streamflow (Webb et al., 2003). Adding such information has the potential to increase accuracy, especially if these factors contain unique information that is unexplained by the spatial correlation of water temperature between sites. For example, if discharge represents a volume of snowmelt water that can influence the correlative relationship of water temperature between sites, inclusion of discharge into the model's structure would increase accuracy.

————————————————————————————————————————

4. Page 6, line 3-5: It appears that the method performs comparably to a linear regression with the exception that the presented method captures extreme thermal pulses and their probability. The linear regression method does not do this. It would be beneficial to emphasize this result and include it in the Abstract.

—[Our reply]—

We fully agree with the reviewer's suggestion. We would modify the text to better stress

out the success in our primary aim recreating extremes:

Results part: We would insert the following sentence in p. 5 line 2 before the sentence starting from "At site B, . . ." "This result indicates that our approach can accurately estimate periodic components and extremes, including the variability in irregularity that cannot be represented by linear regression focusing on an average estimate."

Abstract part: We would insert the following sentence in p. 1 line 16 before the sentence starting from "The results confirm that the developed method. . ." "The result of the performance evaluation indicated that our approach can reasonably estimate periodic components and extremes, including the variability in irregularity, that cannot be represented by linear regression focusing on an average estimate."

————————————————————————————————————————

5. Technical, spelling, and grammatical edits: Page 1, line 10: It would be beneficial to define explicitly what "thermal attributes" are earlier in the manuscript. The authors do so on Page 2, line 21-22. However, the term is used several instances before this definition. Page 1, line 11: "Based on Fourier analysis, this study developed. . ." Misplaced modifier Page 1, line 12-13: "We first quantified. . .Stream temperature was accurately decomposed. . ." The first sentence is active voice while the second sentence is passive voice. The introduction should remain in active voice. Page 2, line 5: Progress in understanding response patterns has been delayed. . ." Subject verb agreement

—[Our reply]—

We would modify these technical corrections accordingly.

————————————————————————————————————————

————————————————————————————

---

## Author Response (AR1)

Dear Editor,

We have modified our manuscript based on the reviewers' comments as follows (note that our replies are represented as ACX.X, following to RCX.X). In addition, we have uploaded the programming scripts which we used in this study as supplement information to enhance reproducibility.

Best regards,

Masahiro Ryo

**Referee #1**

**Replies to RC1**
* * *
**RC1.1.** *Page 2, line 5–6: "Progress in understanding response patterns have been delayed partially because the quantification of thermal attributes is difficult for running waters."*
*The introduction stresses that understanding of thermal attributes in lotic ecosystems is hindered by scarce temporal observations, which certainly is a limitation in current research. However, this method hinges on a reference hydrologic station (i.e. a site that is not data-scarce) to elucidate thermal attributes at a data scarce site. The manuscript's current focus seems to lie in extrapolating within a basin where there is abundant data from at least one location, and it may be helpful to state this explicitly because the description of the limitations in currently available methods/research that are mentioned in the introduction may signal to readers that the proposed method will be rooted solely in spot measurements*

**AC1.1.** As the reviewer has clarified, our approach for temperature estimation based on extrapolation cannot be applied where no monitoring station exists along the streamline. To avoid misleading readers, we have modified the last two sentences in this paragraph and a sentence in the next paragraph according to the reviewer's comment as follows:
* * *
Before (p.2, line 10-): Often in these cases, researchers rely on spot-measures of temperature at study sites and thus lack time-series temperature, thereby limiting understanding of the ecological consequences of thermal attributes in freshwaters. Clearly, an estimate of the thermal attributes at spot-measured sites would benefit this understanding.

After (p.2, line 15-): Often in these cases, researchers rely on spot-measures of temperature at study sites lacking time-series temperature data or refer to temperature time-series data monitored at a nearby hydrological station along the streamline, although likely being biased in thermal attributes. Both datasets have caveats (a lack of time-series or bias in data) when estimating the thermal attributes at a data-poor site, thereby limiting understanding of the ecological consequences in freshwaters. Regardless, estimating thermal stream attributes from both spot-measurements at study sites and time-series measures at the nearest hydrological station would allow more robust estimates.
* * *
Before (p.2, line 15): For instance, regression models employ correlative relationships with air temperature (e.g., Pilgrim et al., 1998) and streamflow (Webb et al., 2003).

After (p.2, line 22-): For instance, regression models employ correlative relationships with air temperature (e.g., Pilgrim et al., 1998) and streamflow (Webb et al., 2003), while a correlative approach considering water temperature at a nearby hydrological station along the streamline have not been implemented to date.
* * *
**RC1.2.** *Page 3, line 5: The authors consider external factors such as weather conditions as part of the irregularity component. However, in other climates, these meteorological conditions would likely factor into the diel (sun) and seasonal (rain) periodicity patterns as well.*

**AC1.2.** For better clarification we have modified the description as follows:

Before (p.3, line 4): We assume that hourly stream temperature $T(t)$ at a given time $t$ is generally composed of the average value $\bar{T}$, seasonal periodicity pattern $S(t)$, and diel periodicity pattern $D(t)$. The remaining component, which is unexplained by these three components, results from multiple external factors such as weather conditions (sunshine, rain, wind, etc.): we call this component an irregularity $\varepsilon(t)$.

After (p.3, line 11-): We assume that hourly stream temperature $T(t)$ at a given time $t$ is generally composed of the average value $\bar{T}$, seasonal periodicity pattern $S(t)$, and diel periodicity pattern $D(t)$. The seasonal and diel periodicity patterns can be driven by meteorological (e.g., solar radiation and precipitation) and hydrological (e.g., discharge and snow-/ice-melt) conditions. The remaining component, which is unexplained by these three components, results from multiple external factors such as stochastic sub-daily changes in weather conditions: we call this component an irregularity $\varepsilon(t)$.
* * *
**RC1.3.** *Page 6, line 5: Are the r2 values (0.66 vs. 0.6) significantly different to state that the proposed method works better than linear regression? It might make more sense to first stress that the method can successfully recreate extremes (cf. linear regression), and then state that it is better at estimating temperatures at site A.*

**AC1.3.** We fully agree with the reviewer's suggestion. We have added the following text to better stress out the success in our primary aim recreating extremes:
(p.6, line 14-): "This result indicates that our approach can accurately estimate periodic components and extremes, including the variability in irregularity that cannot be represented by linear regression focusing on an average estimate."
* * *
**RC1.4.** *Page 7, line 16: Is the threshold value specific to the system in question? How was this value chosen, and what implications might this have?*

**AC1.4.** Yes, it would be system-dependent. The paragraph has been modified by inserting the following sentences. See also newly added supplement figure S1:

(p.8, line 2-): We first estimated an appropriate range of threshold value visually so as to capture a handful of trigonometric curves (i.e., the three major peaks were shown in Figure 3). Then, we compared the patterns modeled based on some threshold values. Too low threshold value results in high sensitivity to noise, while too high threshold value results in high insensitivity to periodic patterns. We compared the performance of the models based on a set of threshold values (0.05, 0.1, and 0.2 °C etc.) and determined graphically as the value 0.1 °C clearly separated periodicity patterns and irregularities (Figure S1; supplement).
* * *
**RC1.5.** *Below is a list of technical corrections:*

*1) "Thermal attributes" should be explicitly described earlier in the introduction. The full description is currently on page 2, line 21, but "thermal attributes" are first mentioned well before this line.*
*2) Page 2, line 26–27: "...multiple scales in time-series data, **and** water temperature in particular."*
*3) Page 2, line 33: "...sites **along** the same stream..."*

**AC1.5.** We have modified technical corrections 1–3 suggested by the reviewer accordingly.
(p.2, line 1): We have described the term at the 1st sentence in the 2nd paragraph, corresponding to *1)*.
We have modified (2; p.3, line 1) and (3; p.3, line 7) accordingly.

**RC3.1.** *Page 2, line 10: To which "problem" are you specifically referring? You could edit this section to read: "...near a hydrological station along the streamline, which is likely biased in thermal attributes. Both limit the understanding of ecological consequences in freshwater."*

**AC3.1.** This point has been also reflected in **AC1.1.**

**RC2.1.** *Page 1, line 16-17: "The results confirm that the developed method can infer stochastic behaviors in stream thermal attributes at spot-measured sites." It would be beneficial to the reader to reword this sentence so that it reflects the requirement of having highly resolved temperature data at a reference site and the assumption of spatial autocorrelation between the reference site and data poor site that this method relies on.*

**AC2.1.** As the reviewer has suggested, we have modified the sentence as follows:

Before (p.1, line 16): The results confirm that the developed method can infer stochastic behaviors in stream thermal attributes at spot-measured sites.

After (p.1, line 20-): The results confirm that the developed method, spatially extrapolating thermal attributes based on Fourier analysis, can infer stochastic behaviors in stream thermal attributes at a data poor site.
* * *
In addition, to increase readability, we described the method more precisely in abstract:

Before (p.1, line 11): This study developed an analytical method to estimate seasonal and diel periodicities as well as irregularities in stream temperature at data-poor sites based on Fourier analysis.

After (p.1, line 11-): This study developed an analytical method based on Fourier analysis to estimate seasonal and diel periodicities as well as irregularities in stream temperature at data-poor sites, by extrapolating thermal attributes from highly resolved temperature data at a reference site on the assumption of spatial autocorrelation.

**RC2.2.** *Page 2, line 3-13: The introduction correctly stresses the importance of knowing "thermal attributes" at a given site with regards to an ecosystem. The authors go on to describe that determining "thermal attributes" can be difficult and unrealistic because of the need for highly resolved temperature data. They present a strong argument for the need for improved modeling that can rely on sparsely collected temperature data. The introduction makes it sound as if the temperature modelling method presented in this manuscript does just that. However, the authors' model is dependent on having two years' worth of hourly temperature data at a reference site. In addition, it relies on the assumption that there is spatial autocorrelation between the reference site and the data poor site. It would be beneficial to reword the introduction so that this information is more explicit.*

**AC2.2.** As the other reviewer has also pointed out the lack of the explanation which can mislead readers (**RC1.1** and **RC3.1**), we have modified the last two sentences in this paragraph and a sentence in the next paragraph (see **AC1.1**).

**RC2.3.** *Page 4, line 9-10: The authors do not include discharge and air temperature data in their methods for simplicity. Would adding this information to the Fourier analysis method improve its performance when compared to the linear regression method?*

**AC2.3.** Yes, adding information on discharge and air temperature has a potential to increase accuracy, especially if these factors contain unique information which is unexplained by the spatial correlation of

water temperatures between sites. For example, if discharge can represent a volume of snow-melting water that may influence the correlative relationship of water temperature between sites, the inclusion of discharge into the model's structure would increase the accuracy.

We have included this point in discussion as follows:

(After p.7, lines 12): As the statistical expression of our approach is linear (Eqs. 1 and 6), it can be easily coupled with approaches in the review; i.e., using other regression models employing air temperature (e.g., Pilgrim et al., 1998) and streamflow (Webb et al., 2003). Adding such information has the potential to increase accuracy, especially if these factors contain unique information that is unexplained by the spatial correlation of water temperature between sites. For example, if discharge represents a volume of snowmelt water that can influence the correlative relationship of water temperature between sites, inclusion of discharge into the model's structure would increase accuracy.
* * *
**RC2.4.** *Page 6, line 3-5: It appears that the method performs comparably to a linear regression with the exception that the presented method captures extreme thermal pulses and their probability. The linear regression method does not do this. It would be beneficial to emphasize this result and include it in the Abstract.*

**AC2.4.** We fully agree with the reviewer's suggestion. We have modified the text to better stress out the success in our primary aim recreating extremes:

Results part: see **AC1.3.**

Abstract part: We have inserted the following sentence in p.1, line 18: "The result of the performance evaluation indicated that our approach can reasonably estimate periodic components and extremes, including the variability in irregularity that cannot be represented by linear regression focusing on an average estimate."
* * *
**RC.2.5.** *Technical, spelling, and grammatical edits:*
*1) Page 1, line 10: It would be beneficial to define explicitly what "thermal attributes" are earlier in the manuscript. The authors do so on Page 2, line 21-22. However, the term is used several instances before this definition.*
*2) Page 1, line 11: "Based on Fourier analysis, this study developed. . ." Misplaced modifier*
*3) Page 1, line 12-13: "We first quantified. . .Stream temperature was accurately decomposed. . ." The first sentence is active voice while the second sentence is passive voice. The introduction should remain in active voice.*
*4) Page 2, line 5: Progress in understanding response patterns has been delayed. . ." Subject verb agreement*

**AC2.5.** We have modified these points accordingly.
1) We have described the term at the 1st sentence in the 2nd paragraph (p.2, line 1).
2) We have modified differently (see **AC2.1 (2)**)
3) We have modified the sentence from passive voice to active voice (p.1, line 14).
4) We have modified it accordingly (p.2, line 10).

[revised manuscript text omitted]

> **Commented [MR13]:** We have attached the programming scripts that were used in this study to enhance reproducibility.

[revised manuscript text omitted]

---

## Author Response (AR3)

Dear Dr. Sally Thompson,

We sincerely appreciate your suggestions on our manuscript. Your comments have substantially improved the presentation of our work. We have revised our manuscript according to all your suggestions. Please confirm our corresponding revisions and replies to your comments.

Best regards, Masahiro Ryo

[revised manuscript text omitted]